# Assessment of Solid Waste Management System in Pakistan and Sustainable Model from Environmental and Economic Perspective

**Asif Iqbal** [1,*] **, Yasar Abdullah** [1] **, Abdul Sattar Nizami** [1] **, Imran Ali Sultan** [2] **and Faiza Sharif** [1]

[1] Sustainable Development Study Center (SDSC), Government College University, Lahore 54000, Pakistan
[2] The Urban Unit, Planning & Development (P&D) Department, Government of the Punjab, Lahore 54000, Pakistan
* Correspondence: asif.iqbal.swm@urbanunit.gov.pk

**Abstract:** The Solid Waste Management (SWM) sector is given a low-priority by the Pakistani Government, with the climate change agenda of Sustainable Development Goals (SDGs) being a priority-3 only, similar to other developing countries. Although sustained efforts have been made during the last decade to strengthen the SWM sector, all actions were focused on manual sweeping and waste collection without integrating waste treatment and disposal options. In this respect, the current model of SWM in the country was analyzed for efficient future planning to strengthen the sector waste management regime in line with the targets of Nationally Determined Contributors (NDCs) and SDGs. An assessment of the SWM sector was performed in eleven major cities of Pakistan, applying Waste-aware benchmarking indicators as strategic tools. The current study highlights the strengths and weaknesses of concerned local municipalities and Waste Management Companies (WMCs) along with interventions to reduce Greenhouse Gases (GHGs) emission targets by 2030. Proposed interventions from the environment and economy perspective will generate revenue to cater for up to 29% of the operational costs, and this will be an important step towards 100% self-sufficiency in the SWM sector.

**Keywords:** SWM system; sustainable waste management; GHGs emission from waste; Pakistan MSW; NDCs; economic model; SDGs; the Urban Unit

## 1. Introduction

Developing countries face many problems, and mismanagement of Solid Waste Management (SWM) leads to low collection efficiency and, resultantly, environmental degradation impacts public health [1]. Developing countries are also facing socio-economic, political, capacity building, institutional and ecological issues with insufficient environmental knowledge that aggregate the sector's unsustainability [2]. SWM is a technical subject with complex systems not conceived by relevant stakeholders in Pakistan. However, waste burning, and open disposal are the main methods to eliminate waste in developing countries that untimely pollute the local environment, negatively impacting the global climate. The waste collection and disposal system can be improved by upgrading the organizational setup with technical interventions [3]. Therefore, the subject should consider integrating with a focus on waste recycling to achieve Sustainable Development Goals (SDGs) targets [4].

Developed countries have achieved targets to improve the SWM sector by implementing legislation that allows diverting waste from landfill to recycling and energy recovery. Municipal Solid Waste (MSW) has the potential to be utilized as a renewable energy source in Europe, and intense competition among two waste treatment options, i.e., incineration and recycling observed, need to be addressed with caution in line with the purview of the environment and circular economy [5]. Long-term sustainability of the SWM sector

required consolation with all stakeholders in a formal working relationship, resource recovery from waste, and implementation of the tipping fee concept to generate funds for investment in current waste treatment facilities, infrastructure development, and public education for waste minimization [6].

Pakistan is the fifth largest country in the world and home to 208 million inhabitants, [7] who produce about 32.6 million metric tons of MSW per annum with an average waste generation rate of 0.43 kg/capita/day [8]. Municipalities manage to collect only 50–60% of generated MSW in Pakistan [9]. The sector's performance regarding service delivery can be gauged and analyzed using different indicators. The main constraints that hinder the progress of the SWM sector include lack of reliable data, inability to design new initiatives in the industry, and to lead the sector towards implementing sectoral policies [10]. The Integrated Waste Management (IWM) approach requires a focus on waste reduction by source segregation and recycling from the recovery of recyclables for an efficient waste management model in low-income cities worldwide [11]. Community engagement is the key to achieving desired results. After recovery of the recyclable materials, residual waste primarily consists of organic components, the relevant authorities must focus on its treatment at this stage, i.e., composting, anaerobic digestion and biogas [12]. Policies on SWM seem unsatisfactory, which calls for the redrafting of existing policies with effective implementation of war-footing. There is a strong desire to strengthen the capacity of relevant authorities in terms of secure budgetary allocation and improvement in current infrastructure for the sector's sustainability [8]. Public Private Partnership (PPP) modality is one mode that can help the desired performance by exploring options for waste-to-energy projects in developing countries [13]. Capacity building of relevant staff, focus on intra-departmental coordination, and public awareness campaigns can help implement SWM policies and action plans in letter and spirit. The success of the SWM sector also depends on priorities set by policymakers [10].

The global quantity of waste disposed of at landfills will be increased by about 3.4 billion tons per annum with a 70% increase in Greenhouse Gases (GHGs) emissions in the year 2050, which is currently 2.01 billion tons per annum. The massive waste disposal is responsible for emitting 11% of global methane into the atmosphere, linked with an increase in the worldwide population [14,15]. Lakhodair, an urban area in Lahore city of Pakistan, has a waste disposal facility with a high emitting source of GHGs with a 12.8% contribution to city-level emissions [16]. The scarcity of available land for waste disposal and allied environmental issues lead to a focus on searching for other alternatives, i.e., available waste treatment facilities with cost recovery options to achieve sustainability in the SWM sector [17].

Waste-aware indicators are a perfect tool for assessing existing SWM systems of the cities as it allows comparing various cities' data to evaluate and implement the strength of a town to overcome gaps in other cities [18]. An assessment of the sector is performed and found that country's SWM system is far behind and needs to define priority areas to track the system towards sustainability. The proposed interventions, i.e., institutional reforms, dedicated collection streams, composting, recycling and integration of the informal sector, will help fill the gaps in Pakistan's current waste management sector; it will also guide the local municipalities, Waste Management Companies (WMCs), and federal and provincial sustainable development support units to design and implement the SWM systems to achieve desired results of SDGs and Nationally Determined Contributors (NDCs). Implementing the proposed interventions will assist in joining the league of upper-middle-class countries by 2030, as the national development plan approved the SDGs of Pakistan in 2016. The initiatives will strike the targets of SDGs [19], i.e., Priority-I; good health and wellbeing, water and sanitation, affordable and clean energy, Priority-II; sustainable cities and communication, partnership for the goals, Priority-III; responsible consumption and production, climate actions and life below water as priorities set by Government of Pakistan [20].

Uncertainties prevail on the part of policymakers for further capital investment in the waste management sector as all WMCs meet the financial requirement from loan money without any cost recovery. Examples include:

- Establishing public sector WMCs across Punjab province;
- Forming Sindh SWM Board;
- Water and Sanitation Services Companies in Khyber Pakhtunkhwa (KPK) province;
- Addition of a new fleet in Quetta, Baluchistan and Punjab;
- Outsourcing waste collection services locally and internationally in various cities.

However, the SWM sector failed to deliver in Pakistan because all efforts were focused on manual sweeping and waste collection, neglecting treatment options. The current study will allow planning in areas requiring administrative reforms with minimal financial investment. The study will also serve as a guideline for the achievement GHGs Emissions Reduction Targets along with revenue generation to lower the financial burden on the government. The role of developed nations is crucial in mitigating global climatic issues. There is an opportunity to pay back to nature by investing in low-income countries like Pakistan for sustainable development with the tactics of offering high prices on landfill methane reduction with technical assistance and green technologies for waste treatment.

## 2. Materials and Methods

Secondary data from 11 cities in Pakistan were collected from concerned municipalities, WMCs, and existing literature on the subject to perform analysis on Waste-aware benchmark indicators, i.e., background information of cities, vital waste-related data of cities, physical characteristics of four major components of MSW and governance factors [21]. In addition, field visits to selected cities were conducted to assess the on-ground quality of services provided by municipalities and WMCs.

### 2.1. Proposed SWM Model for Pakistan

A model for the SWM sector is proposed based on Waste-aware benchmark indicators' evaluation results by evaluating Pakistan's current local conditions and indicators for performance monitoring of the sector, i.e., waste collection efficiency, dedicated waste collection streams, the establishment of waste treatment facilities, i.e., compost and material recovery (MRF), operational arrangements for the transfer station, landfill, and integration of the informal waste sector (see Table 1). Priority areas/scenarios with timelines will escort the policymakers to focus and attract carbon finances from the international market for investment in the SWM sector to achieve targets of revised NDCs [22].

**Table 1.** Proposed SWM model with performance indicators.

| | Performance Indicators | Current Model | Proposed Model |
|---|---|---|---|
| A. | Waste collection efficiency | <75% | ≥85% |
| B. | Waste collection methodology | Single stream | Three dedicated streams |
| C. | Waste transfer | No concept of a transfer station except metropolitan cities have temporary collection points (TCPs) | TCPs for interim arrangement and establishment of transfer station |
| D. | Waste diversion options | Segregation and recycling by the informal sector | Integration of the informal sector with a formal system |
| (a) | MRF | No MRF by municipalities and WMCs. The facility is available at Lahore [1] | MRF with 30% and 50% recyclables |
| (b) | Compost | No composting at present. The facility is available in Lahore [1] | 20% organic waste into compost [2] |
| (c) | Waste disposal | Open dumping and burning | The gas capturing system for old and current disposal sites Utilization of debris and dry sludge for landfill |

[1] Compost and MRF facilities are not operational due to some technical and administrative issues; [2] Compost target set low based on lessons learned from existing facilities.

### 2.2. Environmental Modelling for GHGs Emission

Simulations and GHGs emission model developed by Institute of Global Environmental Strategies (IGES), Japan is used for estimation of GHGs as it is applicable for the Asian Pacific region, including developing countries of the subcontinent. The GHGs emission model is helpful for the calculation and estimation of both emissions, i.e., direct GHGs emission (National Greenhouse Gas Inventory and Carbon Market), and it saves on decision-making [23]. All GHGs emissions and savings are in units; kg of $CO_2$-eq/ton of waste, i.e., mixed recyclables, organic portion, mixed waste, and kg of $CO_2$-eq/month. MSW collected and dumped by municipalities and WMCs is considered for emission estimation, i.e., Gg/year from existing dump sites based on current disposal practices in Pakistan. As per the lesson, 30% of uncollected waste is burned and considered in assumption emissions. The remaining 70% of uncollected waste finds its way into open drains and vacant plots. Four recyclables, i.e., plastic, paper, glass, and metal (Iron and Aluminum) considered for recycling and related analysis. As assumed in modeling, composting from kitchen and green waste is considered with its 100% utilization in agriculture.

### 2.3. Economic Modelling

Recyclables, i.e., paper and cardboard, plastic, glass, and metals (Iron and Aluminum), are considered for recovery from mixed MSW. As per local market surveys, the prices (in Pakistani Rs.) and economic potential for recyclables were determined (see Figure 1). Selling prices for compost are Rs. 8 per kilogram, as per data reported from Lahore Waste Management Company (LWMC). Selling prices per kilogram for recyclables are Rs. 55 for paper and cardboard, Rs. 100 for plastic (average rate for all types), Rs. 3 for glass, and Rs. 250 for metals (Iron and Aluminum).

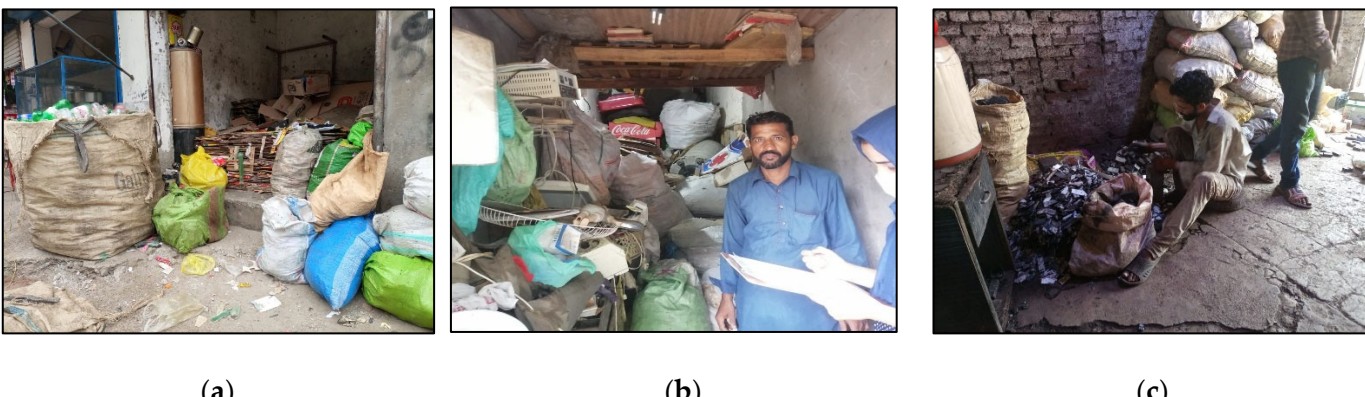

**(a)**                  **(b)**                  **(c)**

**Figure 1.** Survey to determine the economic potential for recyclables: (**a**) Plastic and paper/cardboard; (**b**) Metals and plastic; (**c**) Metals and glass.

Presenting Lahore city as a case study to perform economic modeling of the proposed waste treatment options and related environmental benefits. The idea for economic modeling was perceived from economic assessment [24]. The equations used to perform analysis are as follows:

$$Pd(Fc) = \frac{(Cc - Rc)}{Pl(d)} + \lceil Rc/d \rceil + \lceil Sc/d \rceil + \lceil Mc/d \rceil + \lceil HRc/d \rceil \tag{1}$$

where $Pd(Fc)$ is the per day cost of the facility, $Cc$ is the capital cost to establish the facility, $Rc$ is the residual cost after project life, $Pl(d)$ is project life in days, $\lceil Rc/d \rceil$ is the rental or lease price of land per day, $\lceil Sc/d \rceil$ is the shadow cost, i.e., unseen cost or accidental cost, no objection certificate (NOC), environmental impact assessment (EIA), initial environmental



examination (IEE) cost, etc., per day, $\lceil Mc/d \rceil$ is maintenance cost per day, and $\lceil HRc/d \rceil$ employees or human resource costs per day to operate the facility.

$$Pton(Fc) = \frac{Pd(Fc)}{F\left(\frac{pcap}{d}\right)} \tag{2}$$

where, *Pton(Fc)* is the per ton handling cost of the facility, and *F(pcap/d)* is the waste processing capacity of the facility per day.

$$Pkg(R)Rev = \begin{aligned}&\{Qty.kg/d(p\&c) \times SR(kg)\} + \{Qty.kg/d(psc) \times SR(kg)\}\\&+\{Qty.kg/d(m) \times SR(kg)\} + \{Qty.kg/d(g) \times SR(kg)\}\end{aligned} \tag{3}$$

where, $Pkg(R)rev$ is revenue generated from the sale of recyclables per kilogram, $Qty.kg/d(p\&c)$ is the quantity of paper and cardboard waste per day in kg, $SR(kg)$ is selling prices per kg of the concerned item, $Qty.kg/d(psc)$ is the quantity of plastic waste per day in kilogram, $Qty.kg/d(m)$ is the quantity of metal waste per day in kilogram, and $Qty.kg/d(g)$ is the quantity of glass waste per day in kilogram.

$$Pton(R)Rev = \begin{aligned}&Pkg(R)Rev \div \{(Qty.t/d(p\&c) + Qty.t/d(psc) + Qty.t/d(m)\\&+Qty.t/d(g)\}\end{aligned} \tag{4}$$

where $Pton(R)rev$ is revenue generated from the sale of recyclables per ton, $Qty.t/d(p\&c)$ is the quantity of paper and cardboard waste per day in tons, $Qty.t/d(psc)$ is the quantity of plastic waste per day in tons, $Qty.t/d(m)$ is the quantity of metal waste per day in tons, and $Qty.t/d(g)$ is the quantity of glass waste per day in tons.

$$Pkg(Oc)Rev = Qty.kg/d(comp) \times SR(kg) \tag{5}$$

where $Pkg(Oc)Rev$ is revenue generated from the sale of compost per kilogram, and $Qty.kg/d(comp)$ is the quantity of compost product in kilogram per day.

$$Pton(Oc)Rev = Pkg(Oc)Rev \div Qty.t/d(comp) \tag{6}$$

where $Pton(Oc)Rev$ is revenue generated from the sale of compost per ton, and $Qty.t/d(comp)$ is the quantity of compost product in tons per day.

$$TPton(R\&Oc)Rev = Pton(R)Rev + Pton(Oc)Rev \tag{7}$$

where $TPton(R\&Oc)Rev$ is the total revenue generated per ton from the sale of recyclables, and compost products.

$$CBA\left(\frac{avg}{t}\right) = \begin{aligned}&[\{Pton(R)Rev + Pton(Oc)Rev + Pton(C)Oc\&R(benefit)\} \div 3]\\&-Pton(Fc)\end{aligned} \tag{8}$$

where *CBA(avg/t)* is the average cost–benefit analysis of the project in tons, *Pton(C)Oc&R(benefit)* is carbon finance or environmental benefit from compost and recovery of recyclables per ton.

$$EP(d) = \left\{CBA\left(\frac{avg}{t}\right) \times Qty.\left(\frac{t}{d}\right)Oc\&R\right\} + \left\{Pton(C)lfg(benefit) \times Qty.\left(\frac{t}{d}\right)dw(lfs)\right\} \tag{9}$$

where *EP(d)* is the potential economic value of scenario per day, *Qty.(t/d)Oc&R* is the quantity of compost produced and recyclable recovered per day in tons, *Pton(C)lfs(benefit)* is carbon finance or environmental benefit from landfill gas capturing per ton, *Qty.(t/d)dw(lfs)* is the total quantity of waste diverted at landfill site per day.

## 3. Results

The performance of the SWM sector in service delivery can be gauged and analyzed using different indicators. Waste-aware ISWM benchmark indicators helped to assess

service delivery performance at the city level, decision making and priority setting based on available data; it also highlights the strengths and weaknesses of concerned local municipalities for their focus on targeted planning to fill the identified gaps in the SWM sector [21]. For performance analysis, data were collected from eleven (11) major cities of Pakistan, i.e., Lahore, Karachi, Faisalabad, Gujranwala, Rawalpindi, Multan, Peshawar, Quetta, Hyderabad, Bahawalpur, and DG Khan.

### 3.1. Performance Evaluation of SWM Model in Pakistan

Existing data of cities in Pakistan, including background information about towns, critical waste-related information, physical components of waste, and governance aspects, were obtained and analyzed on waste-aware ISWM benchmark indicators. Selected cities are generating 11.6 million tons of MSW per annum, which comprises 36% of the total waste generation in Pakistan. Cities are chosen from each province of Pakistan to evaluate the provincial priority for the SWM sector. The city sizes vary from urban/megacities, i.e., Karachi and Lahore, where the population is more than 10 million, to a small town/city, i.e., DG Khan, with a population of about 0.4 million (see the location of cities in Figure 2).

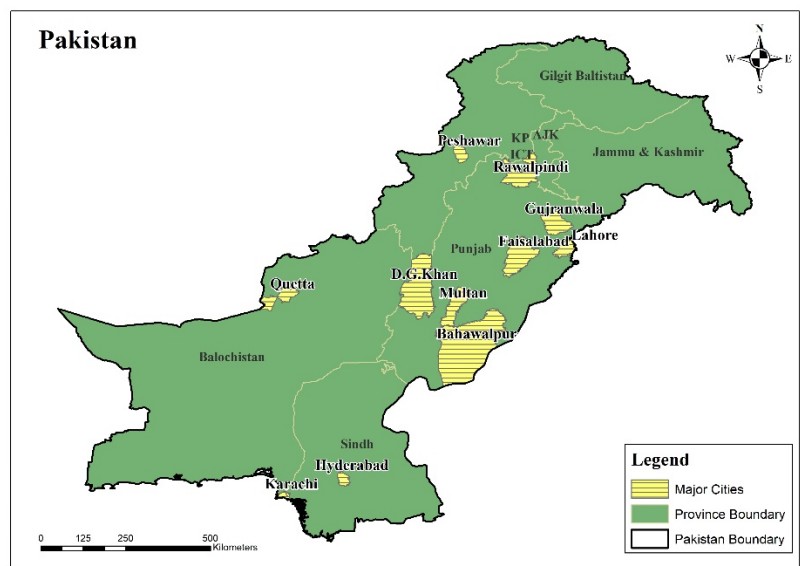

**Figure 2.** Map of Pakistan showing eleven major cities selected to perform SWM model.

### 3.1.1. Background Information and Key Waste Related Data

Pakistan is a low-middle income country with a USD 1500 gross national income [25]. Karachi is the largest city of Sindh, followed by Lahore, the capital and largest city of Punjab. The highest waste generation is observed in Karachi, followed by Lahore, Gujranwala, Faisalabad, Rawalpindi, Quetta, Hyderabad, Multan, Peshawar, Bahawalpur and DG Khan. The city-wise population, waste generation quantity per day and percentage of four major physical waste components are depicted in Appendix A and Figure 3 [9,26–43].

### 3.1.2. Analysis of Physical Components

Data analysis on benchmark indicators (see Appendix A) shows that waste collection coverage varies from 34% for Hyderabad to 90% for Lahore. Faisalabad (43%), Hyderabad (49%), and Gujranwala (34%), where waste captured by the system seems to be >50%. Lahore city ranks in a medium-high category against quality and city cleaning services indicators. Karachi, Faisalabad, Rawalpindi, Gujranwala, Peshawar, Quetta, Bahawalpur, and DG Khan cities ranked in the medium category. Multan and Hyderabad cities ranked low-medium for quality waste collection and street cleaning services. The formation and accumulation of waste heaps, along with waste storage sites, i.e., containers, street corners, etc., are observed in all cities (see Figure 4). Street sweeping frequency varied from the city

center to periphery areas and littering, including vehicle spillage, was observed during waste transportation to TCPs and landfill sites. Low compliance with personal protective equipment, as marked, on the part of sanitation staff (see Figure 4).

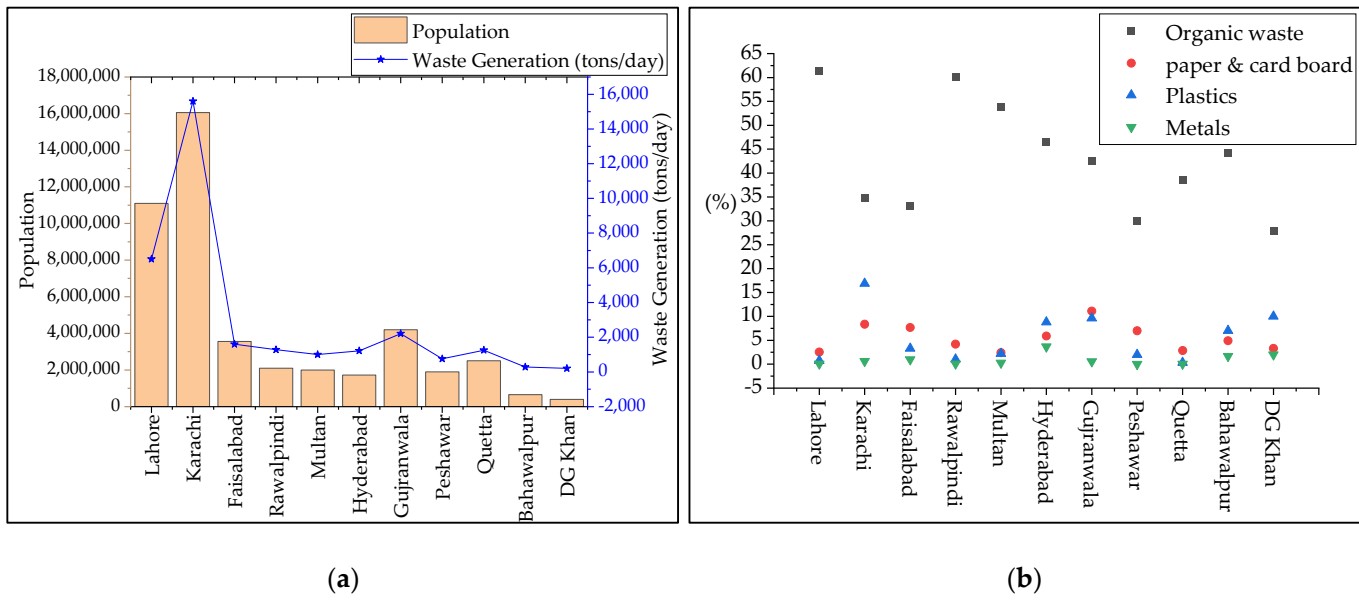

(**a**)　　　　　　　　　　　　　　　　　　(**b**)

**Figure 3.** Population, waste generation trend and percentage of recyclables in selected cities (**a**) Population and waste generation (tons/day) of cities; (**b**) Percentage of 4 major physical components of 11 cities.

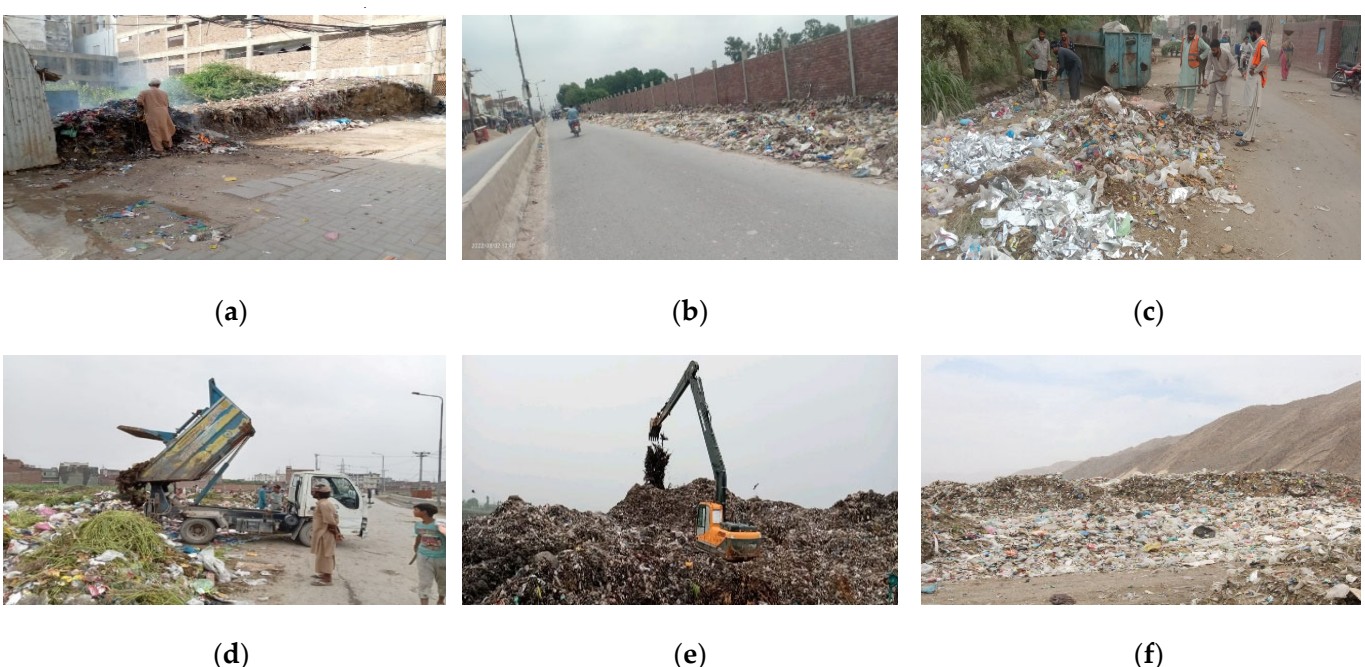

(**a**)　　　　　　　　　　(**b**)　　　　　　　　　　(**c**)

(**d**)　　　　　　　　　　(**e**)　　　　　　　　　　(**f**)

**Figure 4.** Current waste management practices in various cities of Pakistan as (**a**) Burning of waste observed at Latifabad, Hyderabad; (**b**) Open disposal of waste along the roadside, MCT, Multan; (**c**) Waste heaps along storage site and staff working without PPEs at Jaranwala road, Faisalabad; (**d**) Status of open TCP at Gujranwala; (**e**) Status of TCP at Lahore; (**f**) Disposal site of Quetta city.

Poor control of waste treatment and disposal system, as found in Faisalabad (43%), Hyderabad (0%), Gujranwala (34%), and Peshawar (45%) with uncollected waste disposed at illegal sites or remaining scattered on streets which ultimately found its way into wastewater drains. In the absence of any legal framework, the institutions' performance

benchmark cannot measure; resultantly, it is held responsible for deteriorating environmental conditions in the country. In addition, political and economic instability hinders local and international firms from investing in waste treatment technologies. Lahore city with a medium ranking qualified indicator for the quality of the treatment and disposal, i.e., compost and refuse-derived fuel (RDF) plants are installed but, unfortunately, non-operational at the moment due to technical and administrative issues. Other cities found a lack of quality treatment technologies; this indicator is directly linked with sustainable development to achieve the SDGs for Pakistan. Illegal or unauthorized disposal practices (see Figure 4), as observed in most cities, contribute to climate change by the emission of GHGs from unsafe handling of waste [44].

Low effort on recoveries for recyclables was reported on the part of local municipalities and WMCs in all major cities of Pakistan except Lahore, with only 3% recovery of recyclables and organic waste for compost from MSW (currently, the facility is non-operational). Marking and ranking on the recycling rate indicator only represented the efforts made by the informal waste sector, whose actions need municipalities' recognition. Informal recycling rate observed as maximum (26%) in Karachi, which is why the city managed to quality medium category against the indicator. The quality of the reduce, reuse and recycle (3Rs) is also low to low-medium in all cities of Pakistan (see Figure 5; Appendix A).

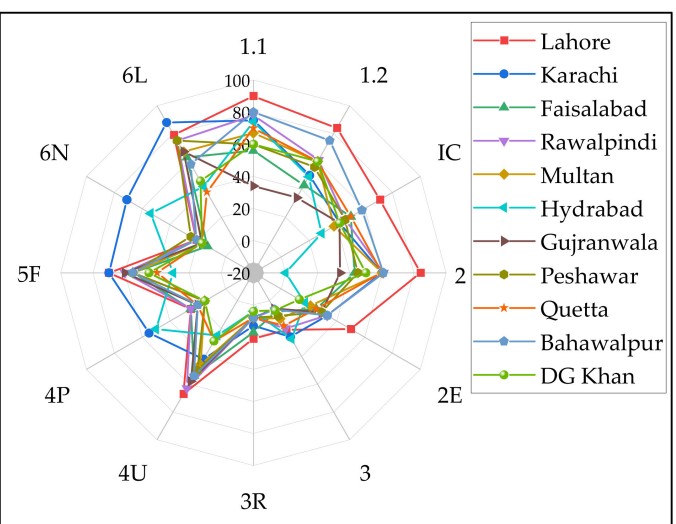

(**a**)

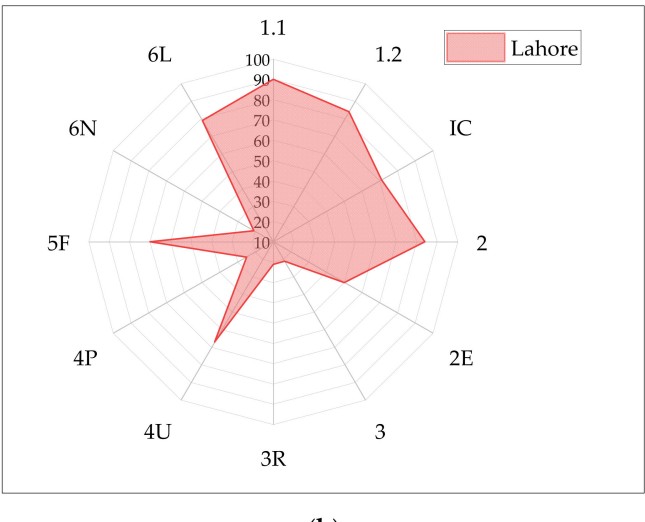

(**b**)

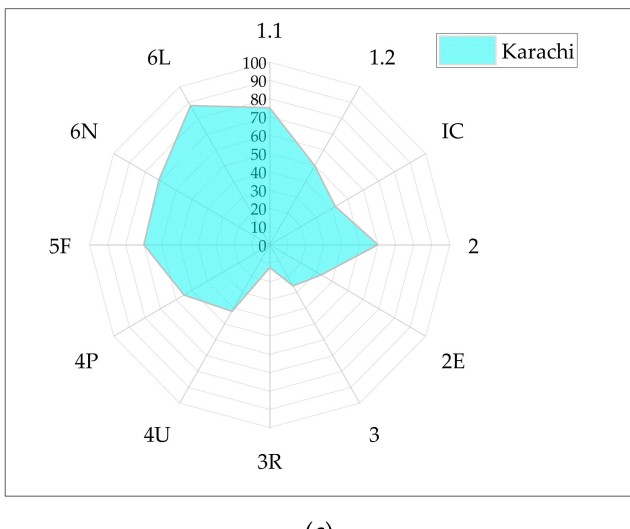

(**c**)

**Figure 5.** *Cont.*

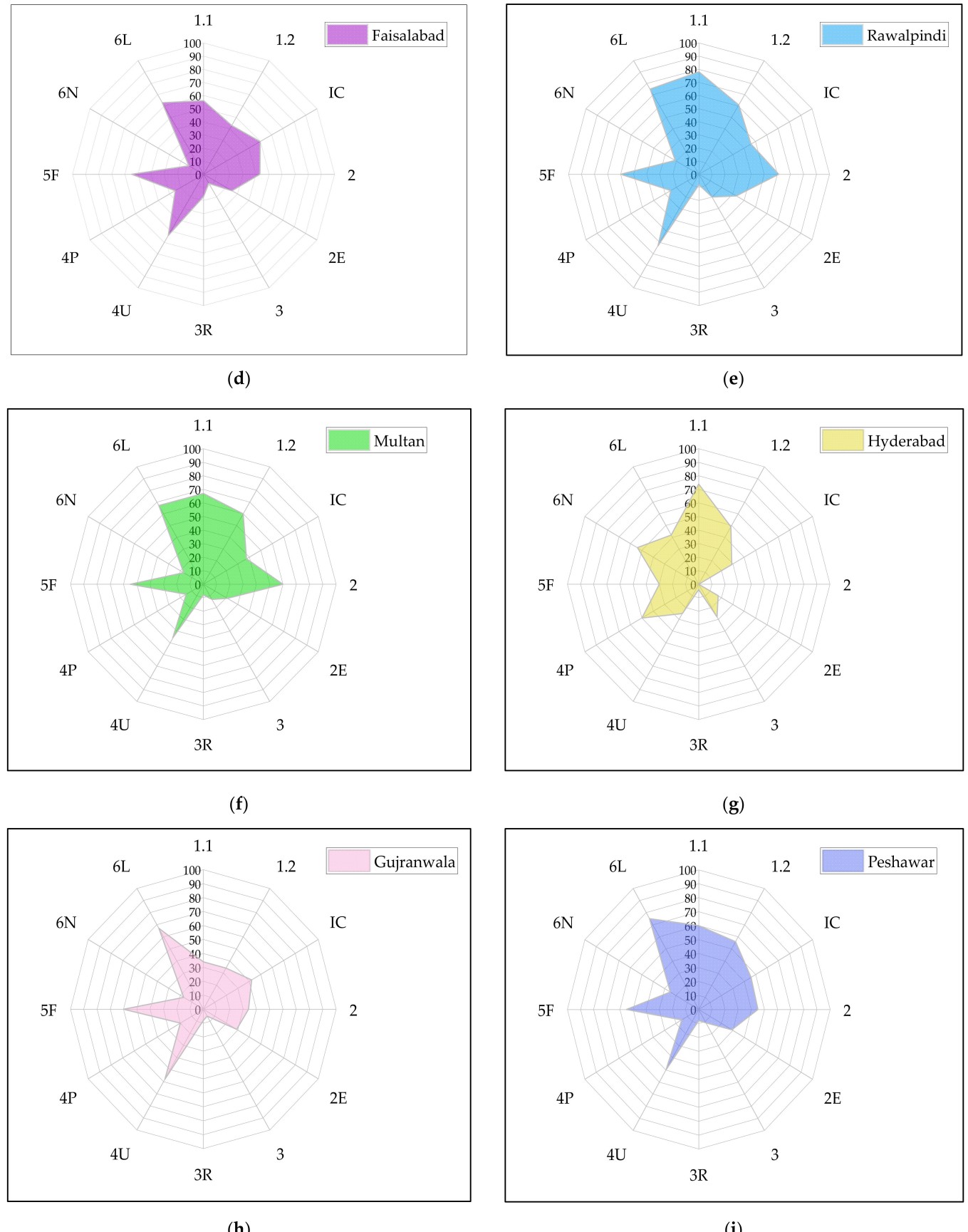

**Figure 5.** *Cont.*

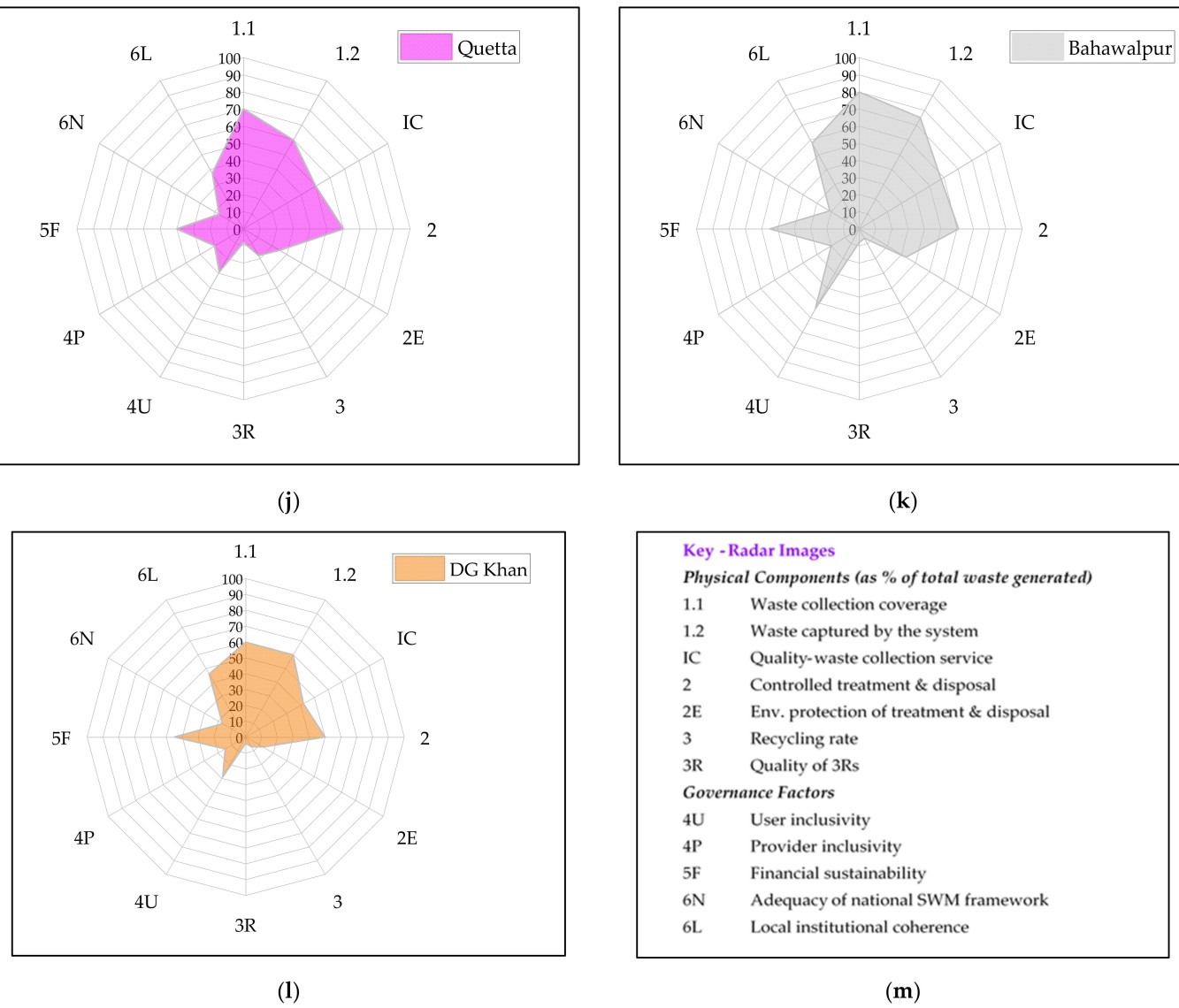

**Figure 5.** (**a**) Comparison of Pakistani cities on waste-aware ISWM benchmark indicators; (**b**) Analysis of Lahore city; (**c**) Analysis of Karachi city; (**d**) Analysis of Faisalabad city; (**e**) Analysis of Rawalpindi city; (**f**) Analysis of Multan city; (**g**) Analysis of Hyderabad city; (**h**) Analysis of Gujranwala city (**i**) Analysis of Peshawar city; (**j**) Analysis of Quetta city; (**k**) Analysis of Bahawalpur city; (**l**) Analysis of DG Kahn city. **Note:** Legend/key of Figure 5 (**a**) to (**l**) is mentioned in Figure 5 (**m**).

### 3.1.3. Evaluation of Governance Factor

Performance analysis for user/provider inclusivity, financial sustainability, institutional capacity, and policy implementation are identified and depicted in Appendix A. Most cities maintained to qualify for medium ranking at inclusive user indicator, i.e., Karachi, Faisalabad, Multan, Gujranwala, Peshawar and Bahawalpur, while Lahore and Rawalpindi ranked in the medium-high category (see Figure 5; Appendix A). The waste collection methodology in these cities is a communal storage container system mostly placed in residential and commercial areas to meet equity of service provision criteria; it was found that all towns have dedicated complaints and helpline numbers, including web pages, to get regular feedback from the public on service delivery. WMCs in Punjab has a dedicated wing for public education and awareness. Hyderabad, Quetta, and DG Khan ranked as a low-medium category for inclusive user indicators (see Figure 5; Appendix A). Consultation with local communities and their involvement in new plans, initiatives, and the right to

be heard from the community was found low due to the absence of any legal obligation in Punjab, Baluchistan and KPK provinces.

Two cities of Sindh province, i.e., Karachi and Hyderabad, ranked as a medium category on the inclusive provider indicator due to the establishment of the Sindh SWM board responsible for policies and integrated approaches for the sector. On the other hand, the remaining cities, e.g., Multan, Gujranwala, Peshawar, Quetta, Bahawalpur, and DG Khan, have ranked as low on inclusive provider indicators. The low ranking was due to the absence of a legal framework in the form of by-laws and SWM act, etc., on the part of the local and provincial governments (see Figure 5; Appendix A). Private sector participation and representation in the SWM sector were found low due to the absence of any law and updated policy on SWM. On the other hand, there is a legally binding obligation for local municipalities and WMCs to follow the rules of the Procurement Regulation Authority (PRA) for hiring and procuring any services and goods. To summarize, transparency in the whole bidding process is ensured in the presence of various watchdogs at different levels.

The financial sustainability aspect is considered crucial for providing regular services for SWM as it is an essential services provision sector in Pakistan. Lahore and Karachi ranked in the medium-high category for financial sustainability indicators, while other cities ranked in the medium category except for Hyderabad and Quetta, which ranked in the low-medium category (see Figure 5; Appendix A). Low cost-recovery was observed and found limited to some commercial entities. Ranking of the cities on the national framework for SWM was observed as low to low-medium except Karachi, which ranked as medium-high category (see Figure 5; Appendix A). Sindh SWM board is responsible for implementing policies, while other provinces are lacking in SWM sector-related legislation, regulations, strategies, and policies with a result framework. Guidelines and implementation procedures were in place in most of the cities of Punjab and the capital cities of Baluchistan and KPK. All the municipalities and WMCs of Pakistan have provisions for enforcement of littering under the local government act; however, implementation is weak.

Most cities ranked medium to medium-high on the local institute's coherence indicator, except Karachi, which qualified to obtain a high rank (see Figure 5; Appendix A). Local municipalities and WMCs have well-established organizational structures with some constraints on the part of institutional capacity; they need to define job descriptions and appoint dedicated human resources for each wing, i.e., enforcement, operations, fleet management, repair and maintenance, and landfill operations. Data related to tonnage, trips, waste characteristics, fleet on-road/off-road status, fuel allocation, deployment of staff, etc. found maintained by WMCs. The federal government can resolve the weak inter-provincial relationships among the municipalities of Punjab, Baluchistan, KPK, and Sindh by assigning the role to specialized institutes at the national level, i.e., the Urban Unit (Urban Sector Planning and Management Services Unit). In addition, Federal Ministry for Planning Development and Special Initiatives, in coordination with Climate Change Ministry, can assign the task to the Urban Unit (UU) for better coordination among provinces. A dedicated "sectorial reform cell" within UU will help to achieve the defined targets of NDCs.

### 3.1.4. Greenhouse Gases (GHGs) Emission from Current Waste Disposal Practices

Emissions of GHGs, i.e., $CH_4$, $CO_2$, and $N_2O$, from solid waste handling are potentially responsible for contributing about 5% of its impact on global climate change [45]. The waste disposal sites around the subcontinent are potentially accountable for emitting 6–26% GHGs into the atmosphere and found that cities are emitting 1.4–2.6 times more than reported emission inventories [16]. The city-level GHGs emissions for Lahore are 50 tons per hour, and the contribution of the Lakhodair disposal facility alone was 13% [16]. There is no single landfill site in Pakistan; hence, all collected waste is dumped openly with some controlled measures by the WMCs. Therefore, estimated GHGs emissions potential from current SWM disposal practices, i.e., open dumping and waste burning, is calculated. The estimated quantity of GHGs emissions from open dumping and garbage burning

(kg. of $CO_2$-eq/ton) for Lahore is 648 and 32; Karachi 535 and 281; Faisalabad 607 and 72; Rawalpindi 684 and 42; Multan 638 and 52; Hyderabad 281 and 148; Gujranwala 621 and 160; Peshawar 358 and 104; Quetta 224 and 42; Bahawalpur 479 and 116; and DG Khan 200 and 110 respectively, as depicted in Figure 6a. The total estimated emission of $CH_4$ from 11 cities in Pakistan is calculated as 179.7 Gg/year, as depicted in Figure 6b.

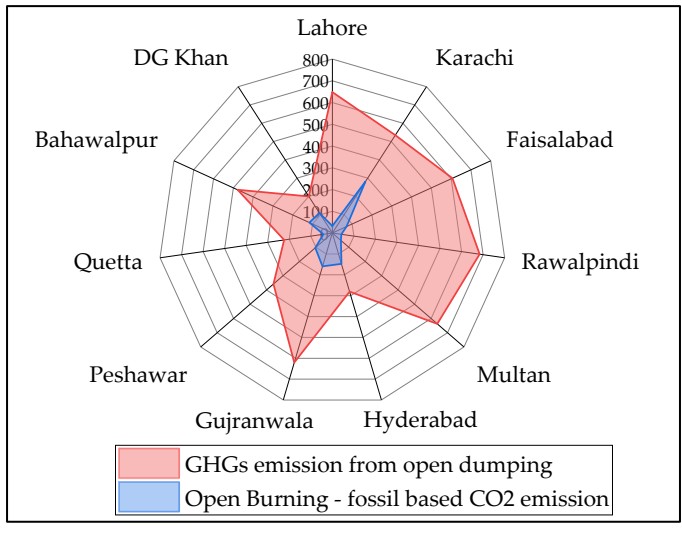
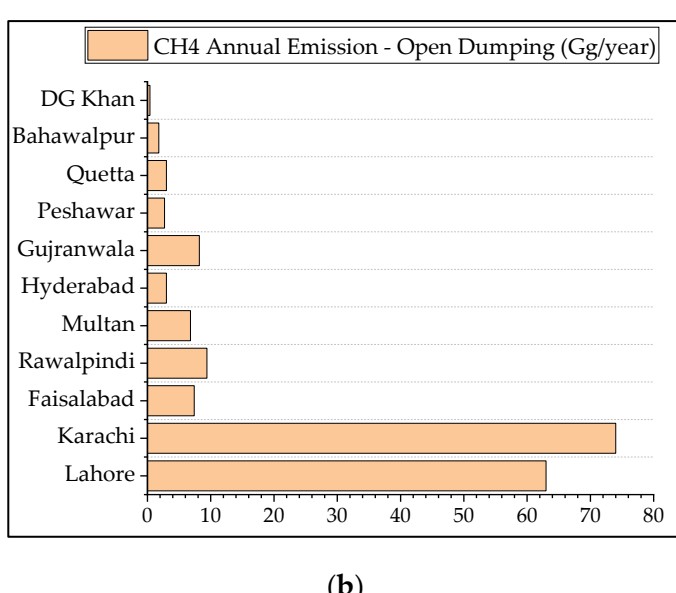

(**a**)                                        (**b**)

**Figure 6.** (**a**) GHGs emission (kg. of $CO_2$-eq/ton of waste) from waste disposal in Pakistan; (**b**) Annual $CH_4$ emission (Gg/year) from dumpsites.

### 3.2. Proposed Sustainable SWM Model for Pakistan

Pakistan is a developing country with an unstable economy and productivity growth, causing a high inflation rate; therefore, the government has to focus more on the current crises in the food and energy sector, which are considered basic necessities for citizens [46]. In such circumstances, the SWM sector is a less-priority area for policymakers and politicians, as evident from the priorities of SDGs. Pakistan needs more focus to improve waste collection efficiency as it found less than 75% in almost all cities except Lahore with 84%. For sectorial sustainability, three separate waste collection streams, i.e., residential, commercial/institutes and bulk waste, are proposed based on the physical characteristics of garbage. The residential waste consists of kitchen waste, raw material for compost manufacturing, and commercial areas' waste is primarily rich in recyclables, requiring a centralized MRF. The facility will help hire the services of the informal sector, i.e., scavengers trained in waste separation, which will help integrate the informal sector into the circular economy. Bulk waste, i.e., debris, will help improve the internal road infrastructure at disposal sites. Recovery of recyclables, compost manufacturing and methane capturing from disposal sites will generate direct revenue for the municipalities. Considering the local socio-economic situation and lessons from the SWM sector, a simple and flexible sustainable model is proposed to increase the sector's efficiency (Figure 7).

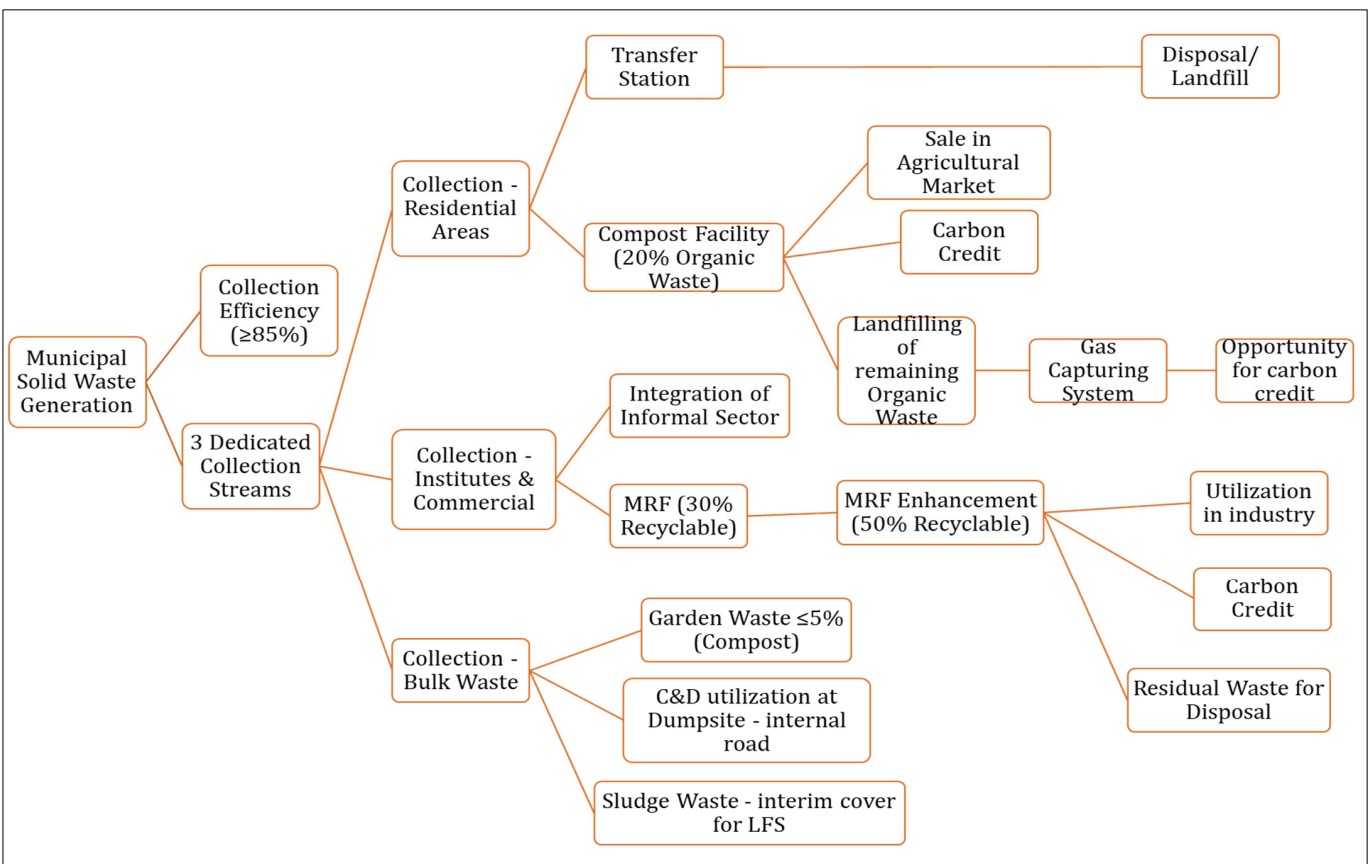

**Figure 7.** Proposed SWM model under local conditions.

The proposed practicable model defines two priority areas, i.e., scenario one and scenario two, with timelines (see Table 2).

**Table 2.** Timelines to implement SWM model to achieve NDCs targets.

| Activities | Year 1 | Year 2 | Year 3 | Year 4 | Year 5 |
|---|---|---|---|---|---|
| Enhancement in collection efficiency (≥85%) | † | † | | | |
| Dedicated waste collection for each stream | | † | | | |
| Arrangements for TCPs | † | | | | |
| Land availability for transfer station and treatment facility | † | † | | | |
| Composting from organic waste (20% organic waste) | | | † | † | † |
| Integration of informal waste sector | † | | | | |
| Establishment of MRF (30% recyclables) | | | ‡ | ‡ | |
| The gas capturing system at the dumpsite | | | ‡ | | |
| Site availability for new landfill | | ‡ | | | |
| Usage of sludge as an interim cover for LFS and utilization of debris waste for infrastructure development | | | ‡ | | |
| Enhancement of MRF facility (50% recyclables) | | | | | ‡ |

Key: † Priority areas (Scenario-1); ‡ Second priority areas (Scenario-2).

The model is compatible with the timespan, i.e., the year 2030, to achieve targets of NDCs for the reduction of GHGs emissions for Business As Usual (BAU). In the priority (Scenario-1), some interventions proposed for policymakers, municipalities, and WMCs that will help to achieve NDCs targets:

- Enhancing the number of trips for existing SWM vehicles, adding some new fleet and equipment, and focusing on repair and maintenance will increase waste collection efficiency minimum at a level of more than 85%;
- Three waste streams, i.e., residential, commercial and institutional, and bulk waste for separate and dedicated waste collection arrangements, will help in quality composting and separation of precious recyclables;
- Identification of state land or procurement of land for waste transfer stations and waste treatment facilities;
- Meanwhile, it is appropriate to have interim arrangements for TCPs per a transfer station's criteria;
- Establishment of the composting facility for 20% organic waste proportion;
- Exploration of the business model of the informal sector for its integration with the formal system.

The second priority areas (Scenario-2) for proposed interventions are as follow:

- Identification and procurement of lands for disposal of waste;
- Establishment of an MRF to cater to 30% of recyclables;
- Arrangements for gas capturing system and flaring of GHGs from current and old dumpsites;
- Utilization of dry municipal sludge as interim soil cover at dumpsites and debris waste for infrastructure development of facilities, i.e., TCPs and disposal sites; Enhancement of MRF facility for recovery of recyclables up to 50%.

The proposed actions are as per their execution types, i.e., administrative, financial, legal, and hybrid. These measures will help the policymakers to define priorities within the sector. As a result, WMCs and municipalities will ensure the sustainability of the SWM sector in the country (see Table 3).

**Table 3.** Actions for the sustainability of the system.

| Activities | Types of Actions Required | | | | Hints |
|---|---|---|---|---|---|
| | Adm. [†] | Fin. [‡] | Legal | Hybrid * | |
| Enhancement in collection efficiency (≥85%) | √ | - | - | - | Focus on Repair and Maintenance (R&M) of fleet and digital monitoring with improved number of trips at disposal sites. |
| Dedicated waste collection for each stream | √ | - | - | - | Reschedule vehicle routes and dedicated vehicles for each stream with color coding and a tracking monitoring system. |
| Arrangements for TCPs | √ | - | - | - | Location in remote areas on a rental basis |
| Land availability for transfer station and treatment facility | - | - | - | √ | Priority to state land, if not available, then acquire land |
| Composting from organic waste ** | - | √ | - | - | Built Operate and Own (BOO) mode with free waste delivery at the facility |
| Integration of informal waste sector, SWM policy, and Act | - | - | √ | - | Legal framework |
| Establishment of MRF facility ** | - | √ | - | - | BOO mode with free waste delivery at the facility |
| The gas capturing system at the dumpsite | - | √ | - | - | Cost-effective solutions |
| Site availability for new landfill | - | - | - | √ | Regional landfill concept on state land |
| Usage of sludge as an interim cover for LFS | √ | - | - | - | Dray sludge waste as interim soil cover. Water and sanitation agencies |
| Enforcement, tipping fee and integration of private entities | - | - | √ | - | Source of revenue generation for WMCs |
| Product stewardship | - | - | √ | - | Source of revenue generation for WMCs |

[†] Administrative action; [‡] Financial action; * Hybrid approach includes administrative, financial and legal actions; ** MRF and compost facilities are considered for capital investment by the government in the absence of external investment to avoid any delay in achieving targets.

### 3.2.1. Environmental Sustainability of Sector

Various waste treatment and technological options are available in the international market. Initially, two possibilities seem feasible to strengthen the SWM sector in Pakistan by focusing on MRF for recyclables and composting for an organic proportion of waste.

### 3.2.2. Recovery of Recyclables

The sustainability of the SWM sector is essential to safeguard the environment. The informal waste sector is considered one of the largest stakeholders in the SWM sector in Pakistan and contributes to economies of scale through informal waste segregation of recyclables and association with the recycling industry. The economic value of the recyclable materials is the driving force in attracting the informal sector to this business. Activities of the current informal sector not only save the transportation cost of the municipality in terms of fuel and increase the waste storage capacity of communal city containers but also reduce operational costs and increase the life of the landfill. The informal sector also contributes to minimizing the environmental hazards by saving emissions of GHGs concerning global climate change, depicted in Figure 8a.

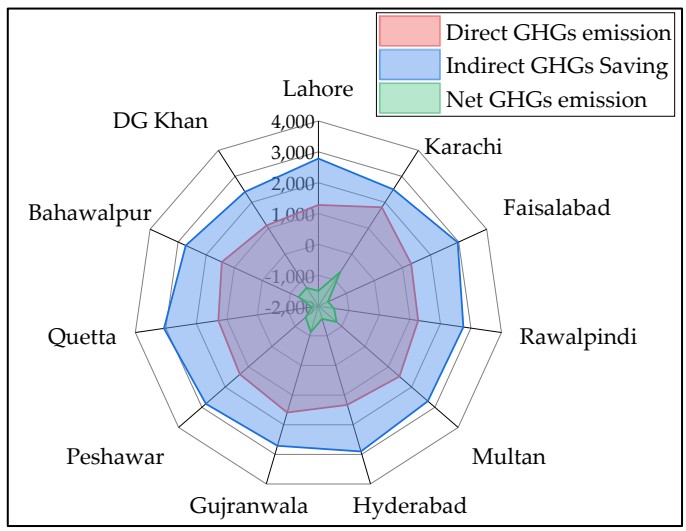 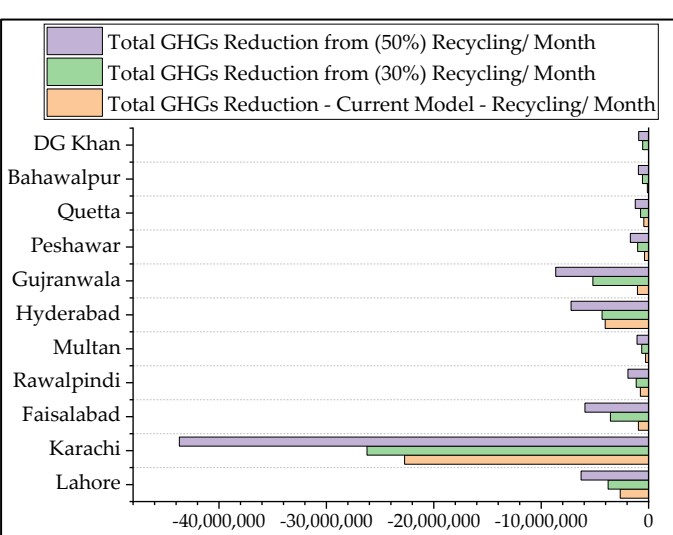

(**a**)          (**b**)

**Figure 8.** (**a**) Saving of GHGs emissions (kg of $CO_2$-eq/ton of recyclables) by informal waste sector; (**b**) Total reduction in GHGs emissions per month (kg of $CO_2$-eq/month) from current and proposed recycling scenarios.

Total GHGs emission reduction is estimated based on three scenarios, i.e., current recycling business by the informal sector, proposed 30% recycling, and 50% recycling targets as depicted in Figure 8b.

The total quantity of recyclable materials for selected cities to be recovered through MRF per month is 53,378 tons against the 30% target (Scenario-1) and 88,968 tons against the 50% target (Scenario-2). Therefore, the city-wise quantity of material per month will be recoverable from mixed waste, as depicted in Figure 9.

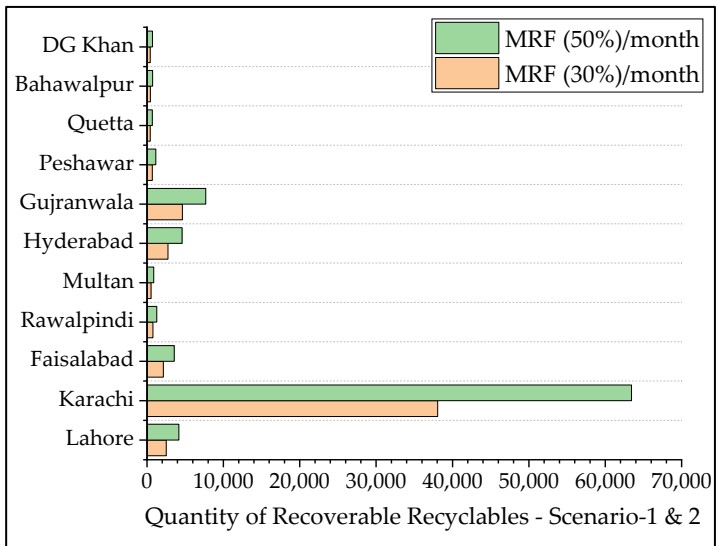

**Figure 9.** Quantity of recoverable recyclables per month against scenation—1 (30%) and 2 (50%).

### 3.2.3. Manufacturing of Compost from Organic Waste

Composting targets apply to those cities that qualify the criteria of waste generation of over 1000 tons per day. Eight cities, i.e., Lahore, Karachi, Faisalabad, Rawalpindi, Gujranwala, Hyderabad, Quetta, and Multan, qualified to meet this criterion. Municipalities and WMCs will be able to initiate environmental and economic sustainability in the SWM sector by executing the plan for compost manufacturing and establishing an MRF. Composting from an organic proportion of municipal waste will save the environment by minimizing net GHGs emissions from a life cycle perspective, i.e., −1306 kg of $CO_2$-eq/ton of organic waste and its further utilization for the agriculture sector. By implementing the proposed model, the SWM sector will be able to sustain the environment in the context of climate change by reducing the total GHGs emissions (see Figure 10a).

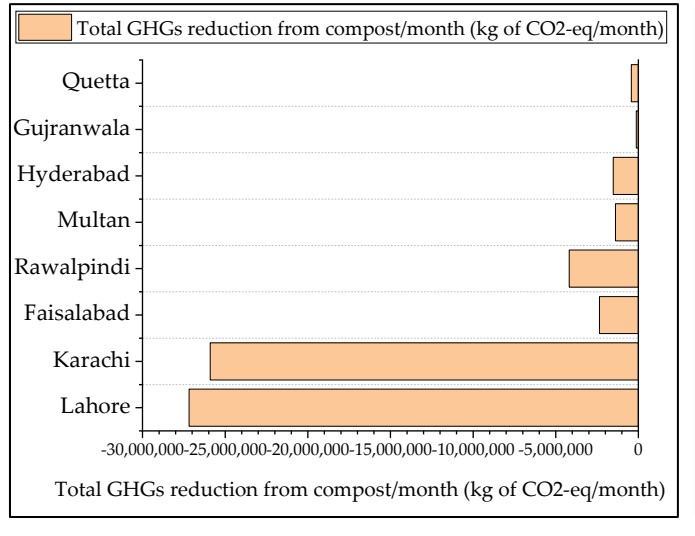

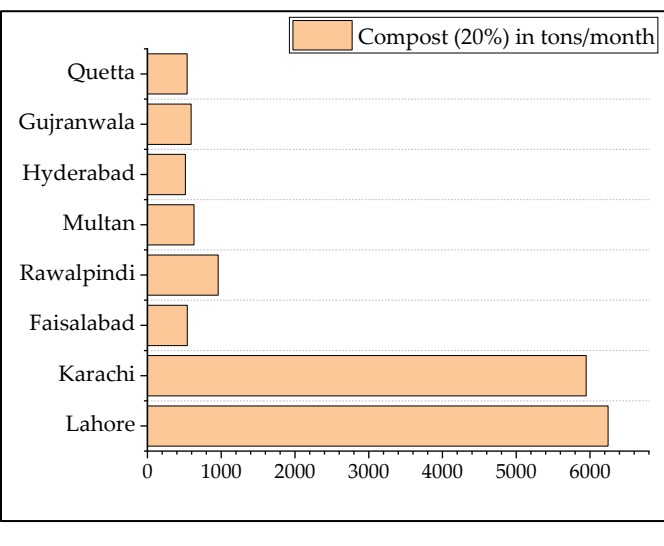

(**a**)                    (**b**)

**Figure 10.** (**a**) City-wise total GHGs reduction (kg of $CO_2$-eq/m) from composting scenario; (**b**) Quantity of compost product (tons/month).

The total quantity of final product, i.e., compost that will recover from organic waste, is calculated as 15,968 tons per month against the 20% target. City-wise compost product amount/month, as depicted in Figure 10b.

### 3.2.4. Economic Sustainability of Sector

Economic analysis of both scenarios was performed; by using Equations (1)–(9). A facility including MRF cum compost plant (1000 tons' waste processing capacity/day) proposed segregating 12.5 tons of recyclables (Scenario-1), 21 tons of recyclables (Scenario-2), and producing 200 tons of compost per day. The facility's capital cost is assumed based on the recently built facility at Sahiwal city. Rs. one billion investments are supposed to establish the facility, including plant infrastructure, access roads, weighbridge installation, offices, windrow pad, leachate pond, etc., assuming plant life as 20 years with a 30% residual value after completion of the project. The land lease amount for a facility is Rs. 24 million per annum. Human resource costs are Rs. 20 million per annum, including engineers, supervisory staff, technicians, sorting labor, windrow labor, etc. For facility maintenance, Rs. 16.8 million per annum is assumed, including fuel cost, electricity expenditure, lubricants, lab equipment, and other admin expenditures. Shadow costs, including license fees, unanticipated costs, and any unseen costs also assumed in the total expenses of the facility. Per ton operational cost for Scenario-1 calculated as Rs. 1350 and Rs. 1299 for Scenario-2 as depicted in Table 4. The cost estimates are for the current year, 2022, and Pakistan's 8.34% average annual inflation rate in the last 20 years needs adjusting per ton costs for future projections.

**Table 4.** Facility establishment and operational cost per ton for both scenarios.

| Cost of Facility | Per Day Cost of Facility | Scenario-1 (MRF 30% and Compost 20%) | Scenario-2 (MRF 50% and Compost 20%) |
|---|---|---|---|
| | Cost (Rs.)/Day * | Cost (Rs.)/Ton ** | Cost (Rs.)/Ton ** |
| Capital investment | 95,890 | 451 | 434 |
| Rent of land | 64,516 | 303 | 292 |
| Shadow cost | 28,200 | 133 | 128 |
| HR to operate the facility | 53,065 | 250 | 240 |
| Maintenance of facility | 45,161 | 213 | 205 |
| Total expenditure of Facility | 286,832 | 1350 | 1299 |

* Equation (1); ** Equation (2).

Expected revenue generation per ton is Rs. 65,949 for Scenario-1 and Scenario-2 from the sale of recyclables and compost (see Table 5).

**Table 5.** Estimated revenue per ton for 2 scenarios.

| Revenue from Sale of Recyclables and Compost | Scenario-1 (MRF 30% and Compost 20%) | Scenario-2 (MRF 50% and Compost 20%) |
|---|---|---|
| | Revenue (Rs.)/Ton | Revenue (Rs.)/Ton |
| Revenue from the sale of recyclables * | 57,949 | 57,949 |
| Revenue from the sale of compost ** | 8000 | 8000 |
| Total estimated revenue (Rs.) *** | 65,949 | 65,949 |

* Equation (4); ** Equation (6); *** Equation (7).

Carbon credit pricing for landfill methane [47] ranges from USD 0.2–19 for $1MtCO_2$-eq and an average price of USD 2 (Rs. 224.35 = USD 1) taken for carbon credit (see Table 6).

**Table 6.** Environmental benefit revenue.

| Revenue from Carbon Credit/Environmental Benefit | Scenarios 1 and 2 |
| --- | --- |
| | Revenue (Rs.)/Ton |
| Carbon benefits from recyclables | 673 |
| Carbon benefits from compost | 583 |
| Carbon Benefit—Gas capturing from the dumpsite | 287 |
| Total revenue (Rs.) carbon credit | 1543 |

The total expected revenue generation from the sale, i.e., recyclables, compost, and related benefits from carbon credits, are depicted in Table 7.

**Table 7.** Average cost–benefit analysis per ton.

| Cost–Benefit Analysis * | Scenario-1 | Scenario-2 |
| --- | --- | --- |
| | Revenue (Rs.)/Ton | Revenue (Rs.)/Ton |
| Per ton revenue from the sale of recyclables, compost and relevant carbon benefit | 22,402 | 22,402 |
| Per ton facility cost | −1350 | −1299 |
| Average per ton cost–benefit analysis | 21,052 | 21,103 |

* Equation (7).

Economic potential per day for Scenario-1 and Scenario-2; was determined as Rs. 4,472,455 and Rs. 5,939,161, respectively, per Equation (8). The overall economic potential of the proposed model (see Table 8) showed that WMC at Lahore could sustain the SWM system by catering to its operational costs of up to 29% of revenue money.

**Table 8.** Economic benefit of the proposed model.

| Scenarios | Qty. of Compost and Recyclables/Annum | Economic Potential (Rs. Million)/Annum * | Operational Cost (Rs. Million) of LWMC/Annum | % (Revenue Benefit)/Annum to Cater Operational Cost |
| --- | --- | --- | --- | --- |
| Scenario-1 | 67,984 | 1431 | 6500 | 22 |
| Scenario-2 | 70,640 | 1901 | 6500 | 29 |

* Annual economic potential calculated on 320 days.

## 4. Discussions

After detailed analysis, some areas are "highlighted" for waste collection, treatment, disposal, and resource management improvement. The results obtained from this study will allow the policymakers to prioritize investments in identified areas that may hinder the performance and achievement of desired goals of SDGs for Pakistan [48]. The budget allocated to WMCs in Punjab is loan money which is not a sustainable solution in the long term. Waste collection/operational cost per ton varied from the highest, i.e., Rs. 7737 for Rawalpindi, to the lowest, i.e., Rs. 2905 for Bahawalpur [49]. The average waste collection cost per ton is Rs. 4794 for urban areas of Pakistan. Proposed sustainable interventions will help cater to the per ton operational cost, i.e., Rs. 1088, by benefiting in terms of revenue generation. Lahore is the only city that recovered minimal cost from the households via the Water and Sanitation Agency billing system operating locally in the area jurisdiction of ex-Municipal Corporation Lahore (with 65% area coverage). The municipalities did not conceive the concept of tipping fees due to the unavailability of authorized disposal sites in most of the cities of Pakistan. Lahore is the only city that introduced the tipping fee concept in Pakistan by notifying Standard Operating Procedures (SOPs) in consultation with the Lahore Development Authority (LDA). Some percentage of the mortgaged property is linked with clearance/NOC from LWMC for the development of new housing societies in the area jurisdiction of Lahore district.

Current waste disposal methods, i.e., open dumping and waste burning, are considered significant GHG emission sources while handling solid waste that contributes to climate change [50]. About 30% of uncollected waste burns by the municipality's sanitary workers/gardeners and the local community. The emission of total particulate matter (TPM) from the burning of MSW is composed of $PM_{10}$ and $PM_{2.5}$ at 80% and 52% [51], respectively, which degrade the ambient air quality and affect public health [52]. Lahore declares the "most polluted city" worldwide [53], and waste burning is one factor contributing to the city's smog issue [54]. The waste sector is included in revised NDCs for Pakistan, with 50% emission reduction targets by 2030 [55]. The industry has excellent potential for its alignment with low-emission waste treatment technologies [56]. Investment in the sector by donor agencies in the form of carbon financing and green bonds will not only strengthen the capacity of local municipalities and WMCs but also help mitigate global climatic issues. The first area for investment in the sector is compost manufacturing from organic waste. MSW of Pakistani cities found rich in organic components and suitable for composting that will have the potential to utilize in the agriculture sector and gardening at the city level by the concerned municipality itself, i.e., Parks and Horticulture Authority (PHA) or wing of a municipal corporation operating in various urban cities for the management of public gardens, green belts, and landscaping. The second area for investment in the sector is the recovery of recyclables from waste; therefore, the MRF facilities need to establish to achieve targets. These two interventions will be a significant waste diversion from open dumping and, in return, will save the environment by lowering up to 79,579 tons of $CO_2$-eq/month for recycling and 63,099 tons of $CO_2$-eq/month for compost from selected cities. There is only a need to explore further the business model of the informal sector, its possibility, and the potential for integration with the formal SWM sector to achieve the self-sufficiency of the system. A third central area for investment is to rehabilitate old and current waste disposal sites in Pakistan. Installation of gas collection [57] infrastructure at disposal sites will help to capture GHGs for further utilization as energy. The Punjab government has recently established a PPP authority at the provincial level, which provided a platform for external investment, i.e., an opportunity for the private sector to invest in any type of project, including the SWM sector. The proposed three primary interventions will help reduce the government's financial burden. Moreover, the final results will prove as a benchmark that will guide the local municipalities, WMC, politicians and international sectoral investment/donor agencies to invest in SWM-related infrastructure [58] in developing countries to save the global climatic calamities.

Local industry for fabrication of SWM-related equipment and machinery has developed well in the country during the last decade, and credit goes to international outsourcing of waste collection services in mega cities of Pakistan. Implementation of development, enhancement, embedding, defense and corrective strategies by the policymakers to help improve the country's poor waste treatment and disposal practices [59]. However, there is a dire need to focus on new waste collection models to enhance the efficiency of waste collection and its integration with treatment facilities. For example, Walton Cantonment Board (WCB) Lahore has recently taken a "Zero-cost initiative" to recover recyclables in their jurisdiction; it is the first kind of initiative in the formal government sector. The WCB piloted a project in the Askari Housing Scheme for source-level segregation of recyclables, i.e., paper and cardboard, polythene bags and metal, glass and plastic. The sanitary inspectors of the municipality provided the residents with source segregation training. Response from the community is very positive, and people are now well-trained in source segregation. WCB deployed a modified vehicle with three color-coded compartments, i.e., White for polythene bags, Yellow for paper and cardboard and Blue for metal, glass and plastic (see Figure 11). Segregated waste is then hauled and stored at a warehouse with dedicated arrangements for storing each segregated component, as depicted in Figure 11b. Recyclables sell to the recycling industry, generating revenue for the municipality workers to motivate the staff towards such interventions. Such initiatives will help the sector's sustainability and set a direction for other municipalities for such interventions. There is

only a need to regulate waste management under one umbrella, i.e., establishing SWM authorities at the provincial level with dedicated enforcement and revenue collection powers to streamline the sector in Pakistan.

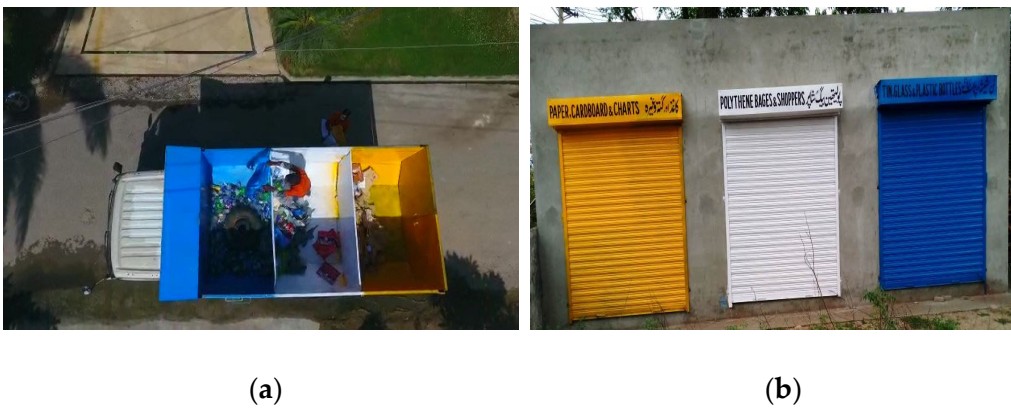

|  (a)  |  (b)  |

**Figure 11.** (**a**) Modified vehicle for recyclables collection; (**b**) Warehouse for each recyclable.

## 5. Conclusions

Analysis of the cities' data at Waste-aware ISWM benchmarking indicators [60] helped identify the gaps for inefficient service delivery in Pakistan. The study's objective was to provide a practical solution to policymakers under uncertain local conditions to focus on targeted planning for the sustainability of the SWM sector [61] in Pakistan. Local governments should draft the SWM Act and related policy in consultation with the Ministry of Climate Change for its implementation in all provinces. Organizational restructuring of the SWM sector is essential to transform company mode to provincial authority, and engagement of the private sector must be encouraged in waste treatment for advanced environmentally friendly technologies. There is also a need at the international level to offer high carbon prices for landfill methane reduction to low-income countries, which will help attain sustainability in the industry under the Paris agreement's prerogative [62]. The enormous potential is available to reach environmental sustainability by diverting waste from landfills, considered the cheapest waste disposal method [59]. However, municipalities in low-income countries need help to calculate the resources for waste collection and haulage due to capacity issues, and future research needs to design a new model/calculator for the ease of municipalities to sustain the sector.

**Supplementary Materials:** The following supporting information can be downloaded at: https://www.mdpi.com/article/10.3390/su141912680/s1, Suplementary Excel.

**Author Contributions:** A.I. contributed to writing the original draft; I.A.S., Y.A., A.S.N. and F.S. contributed to revision and final editing; All authors have read and agreed to the published version of the manuscript.

**Funding:** This research received no external funding.

**Institutional Review Board Statement:** Not applicable.

**Informed Consent Statement:** Informed consent was obtained from all individuals surveyed in the study to use their pictures and data collected from local municipalities and WMCs.

**Data Availability Statement:** Data is available in Supplementary Materials.

**Acknowledgments:** The author appreciates Waste-Aware Integrated Solid Waste Management (ISWM) benchmark indicators that helped developing nations like Pakistan to analyze and assess the current SWM system for better planning of the sector. The author also highly appreciates the support of the Institute of Global Environmental Strategies (IGES), Japan, for providing a GHGs emission model to perform an environmental analysis of the sector. The support of Ghulam Mohey-ud-Din, Urban Economist at the Urban Unit, is also acknowledged for his guidance in the study.

**Conflicts of Interest:** The authors declare no conflict of interest.

## Appendix A. Wasteware ISWM Benchmarking Indicators—Assessment of Eleven Major Cities of Pakistan

| No | Category | Indicators | Results | | | | | | | | | | |
|---|---|---|---|---|---|---|---|---|---|---|---|---|---|
| | | Cities | Lahore | Karachi | Faisalabad | Rawalpindi | Multan | Hyderabad | Gujranwala | Peshawar | Quetta | Bahawalpur | DG Khan |
| **Background information on the city** | | | | | | | | | | | | | |
| B1 | Country income level | World Bank income category | Lower middle income | | | | | | | | | | |
| | | GNI per capita | $1,270 | | | | | | | | | | |
| B2 | Population | Total population of the city (million) | 11.10 | 16.05 | 3.56 | 2.10 | 2.00 | 1.72 | 4.20 | 1.90 | 2.50 | 0.65 | 0.40 |
| B3 | Waste generation | MSW generation (tons/day) | 6,500 | 15,600 | 1,600 | 1,280 | 1,000 | 1,213 | 2,208 | 753 | 1,250 | 282 | 200 |
| | | MSW generation (tons/year) | 2,372,500 | 5,694,000 | 584,000 | 467,200 | 365,000 | 442,745 | 805,920 | 274,845 | 456,250 | 102,930 | 73,000 |
| **Key Waste-related data** | | | | | | | | | | | | | |
| W1 | Waste per capita | MSW per capita (kg/day) | 0.54 | 0.76 | 0.45 | 0.61 | 0.48 | 0.6 | 0.5 | 0.4 | 0.5 | 0.42 | 0.5 |
| | | MSW per capita (kg/year) | 197 | 277 | 164 | 223 | 175 | 219 | 183 | 146 | 183 | 153 | 183 |
| W2 | Waste Composition: | | | | | | | | | | | | | |
| W2.1 | Organic | Organic (food & green waste) | 61.31 | 34.84 | 33.18 | 60.13 | 53.87 | 46.51 | 42.49 | 30 | 38.6 | 44.33 | 28 |
| W2.2 | Paper | paper & card board | 2.53 | 8.36 | 7.67 | 4.2 | 2.4 | 5.89 | 11.12 | 7 | 2.87 | 4.9 | 6 |
| W2.3 | Plastics | Plastics | 0.74 | 16.86 | 3.3 | 1.04 | 2.2 | 8.79 | 9.62 | 2.0 | 0.34 | 7.01 | 10 |
| W2.4 | Metals | Metals | 0.09 | 0.61 | 1.00 | 0.09 | 0.3 | 3.66 | 0.56 | 0 | 0.05 | 1.69 | 2 |
| **Physical Components: 4 key fractions - as % of total waste generate** | | | | | | | | | | | | | |
| 1.1 | Public health - waste collection | Waste collection coverage | 90 | 75 | 56 | 78 | 67 | 74 | 34 | 60 | 70 | 80 | 60 |
| 1.2 IC | | Waste captured by the system | 84 | 50 | 43 | 61 | 60 | 49 | 34 | 56 | 60 | 75 | 60 |
| | | Quality-waste collection service | 71 | 42 | 50 | 46 | 38 | 29 | 42 | 46 | 50 | 58 | 42 |
| 2 | Environmental control - waste treatmnet & disposal | Controlled treatment & disposal | 84 | 60 | 43 | 61 | 60 | 0 | 34 | 45 | 60 | 61 | 50 |
| 2E | | Env. Protection of treatmnet & disposal | 50 | 33 | 25 | 33 | 21 | 17 | 29 | 29 | 25 | 33 | 13 |
| 3 | Reduce, reuse & recycle | Recycling rate | 21 | 26 | 8 | 20 | 13 | 28 | 6 | 11 | 18 | 6 | 7 |
| 3R | | Quality of 3Rs | 21 | 13 | 17 | 8 | 8 | 4 | 8 | 8 | 8 | 8 | 4 |
| **Governance Factors** | | | | | | | | | | | | | |
| 4U | Inclusivity | User inclusivity | 67 | 42 | 54 | 63 | 46 | 25 | 58 | 50 | 29 | 54 | 29 |
| 4P | | Provider inclusivity | 25 | 55 | 25 | 25 | 15 | 50 | 20 | 15 | 20 | 20 | 15 |
| 5F | Financial sustainability | Financial sustainability | 70 | 70 | 55 | 60 | 55 | 30 | 60 | 55 | 40 | 55 | 45 |
| 6N | Sound institutions, proactive polices | Adequacy of national SWM framework | 21 | 71 | 13 | 21 | 17 | 54 | 17 | 25 | 17 | 21 | 17 |
| 6L | | Local institutional coherence | 79 | 88 | 63 | 75 | 67 | 42 | 67 | 75 | 38 | 58 | 46 |

**Key for color coding**

| | |
|---|---|
| Low : Red | |
| Low/ Medium: Red/ Orange | |
| Medium: Orange | |
| Medium/ High: Orange/ Green | |
| High: Green | |

**Key for Abbreviations**

| | |
|---|---|
| B – Background Data | 4U – User Inclusivity |
| W – Waste Data | 4P – Provider Inclusivity |
| 1C – Public Health | 5F – Financial Sustainability |
| 2E – Environmental Co... | 6N – National Framework |
| 3R– Resource Managem... | 6L – Local Institution |

**Figure A1.** Pakistan cities data is incorporated in a table as per instruction described in the user manual 'Waste-Aware' Benchmark Indicators for Integrated Sustainable Waste Management in Cities [21] and Waste-Aware Benchmark Indicators for Integrated Sustainable Waste Management in Chinese Cities [60].

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
