# Peer review of "Assessment of Solid Waste Management System in Pakistan and Sustainable Model from Environmental and Economic Perspective"

_sustainability, doi:10.3390/su141912680_

Round 1

Reviewer 1 Report

This study focused on solid waste management in Pakistan. The strengths and weaknesses of concerned local municipalities and waste management companies were discussed aiming to reach strategies to reduce greenhouse gases emission targets by 2030. This research is considered a case study, The following points should be taken into account to improve the current manuscript:

1- This is a critical point, the authors used many references, but I cannot observe any of its citations within the text, the authors must rewrite the manuscript to take this point into consideration 

2- Let readers know what others have done in the field of solid waste management and bring out clearly the gap left that you want to fill in this study in the Introduction part.

3- The conclusion part is too long

4- Page1, Lines 32-33, and Page2, Lines 62-64, What is the source of this information? 

5-Add sources for equations 1-6

6- Line 146, do you mean Sr or SP, Please correct

7-  Please add an informative title for Figure 2

8-Figure 3 (b), add a y-axis label.

Author Response

The following points should be taken into account to improve the current manuscript:

Point 1: This is a critical point, the authors used many references, but I cannot observe any of its citations within the text, the authors must rewrite the manuscript to take this point into consideration 

Reply: The citation of references is mentioned within the text properly. Previously, it was missed due to some software error when the document was converted from word to PDF format.

Point 2: Let readers know what others have done in the field of solid waste management and bring out clearly the gap left that you want to fill in this study in the Introduction part.

Reply: The introduction part is updated by incorporating the observations, i.e., what others did in lines 47-56; gaps left in lines 59,60, 34-43; and filling the gaps in lines 89-100.  

Point 3: The conclusion part is too long

Reply: Authors tried their best to shorten the conclusion.

Point 4: Page1, Lines 32-33, and Page2, Lines 62-64, What is the source of this information? 

Reply: Lines 32-33 are updated as 57-59 with references. Lines 62-64 are updated as 79-83 with references. 

Point 5: Add sources for equations 1-6

Reply: Equations are developed by the authors by perceiving idea as mentioned in lines 167-169 with reference.

Point 6: Line 146, do you mean Sr or SP, Please correct

Reply: It's SP. Thanks a lot for the correction. The author really appreciates and is proud to know that a real expert is on it.

Point 7: Please add an informative title for Figure 2

Reply: added, please.

Point 8: 8-Figure 3 (b), add a y-axis label.

Reply: Y-axis labelled, please.

As per observations in the Reviewer Table; the authors tries best to improve the crossed points.  

Reviewer 2 Report

Iqbal et al. studied Solid Waste Management System in Pakistan by Applying Waste-aware Benchmark Indicators and Sustainable Model from Environmental and Economic Perspective. The manuscript has certain innovations, but there are problems in the following aspects:

1. The language of the manuscript needs to be greatly improved.

2. Carefully check all abbreviations in the manuscript.

3. The abstract section is too long and the author is advised to simplify it to a certain extent.

4. The introductory part is unattractive and the author is advised to give a more detailed description.

5. The format of the reference is suggested to be revised by the author referring to the standard of the journal.

Author Response

there are problems in the following aspects:

Point 1: The language of the manuscript needs to be greatly improved.

Reply: The authors tried their best to improve the language in an updated version which is available in track changes.

Point 2: Carefully check all abbreviations in the manuscript.

Reply: compliance ensured, please.

Point 3: The abstract section is too long and the author is advised to simplify it to a certain extent.

Reply: Abstract section is updated as per the journal's requirement. 

Point 4: The introductory part is unattractive and the author is advised to give a more detailed description.

Reply: Introductory part is revised as per comments and can be viewed by track change. An effort is made to make it more attractive. Thanks for the kind suggestion.

Point 5: The format of the reference is suggested to be revised by the author referring to the standard of the journal.

Reply: Yes, there was a real need to update the references in line with the journal format. Compliance ensured with thanks. 

Reviewer 3 Report

In this study, authors have focused on the Assessment of Solid Waste Management System in Pakistan. This manuscript shows some interesting findings regarding the SWM sector in Pakistan which was performed in eleven major cities of Pakistan, applying waste-aware benchmarking indicators as strategic tools. Overall, the manuscript is good constructed, understandable and with enough literature review to meet the requirements of the journal.

Specific comments:

1.     Authors are advised to check the whole manuscript for typograph errors. Example: some of the sentences at the end have a space before point. Page 1, line 33; 38;46, page 2, line 48; 64;94, page 6, line 179; 186;189;194;197;201;204; 209; 212.......

2.     Some of the abbreviation are not defined for the first time.

3.     Page 1, line 43:“Policies on SWM seem satisfactory, and there is a need for implementation“. If they seem satisfactory, why is then a need for implementation?

4.     Page 2 line 72:“The uncertainties observed on the part of policymakers for further capital investment in the sector because of lessons learned during the last decade.“ Unclear? Revise.

5.     Table 1. Align text and labels

6.     Authors are advised to centre all formulas/Equations and to crossref. with the text. Also attention should be directed to equation 3. There should be a large bracket at start and before:1000, in the situation as it is written it means that only the last bracket: (Qty*kg/d(g)xSP(kg)) is divided by 1000. As I understood all brackets need to me calculated as SUM, then divided by 1000 to get the result in recyclables per ton.

7.     Page 6, line 182 authors write: paper 6%, and if we look at the Appendix for paper is 8.36% stated. Which one is correct? Please check.

8.     Authors should uniformly state results for same parameter. Example: 6%, 1,8%, 60.13%....same parameter (%), but with different decimal places??

9.     Numbers are sometimes written by point and sometimes by comma. Correct, to be uniformly.

10.   Figures should also be consistent, with same style. Also the text should be same as in other parts of MS. Times New Roman.

11.  Re-check page 7, line 223-226 with Appendix values. Line 236- Peshawaer (45%) is not visible in the Appendix section?

12.  Some sentences have the same meaning, which means that some obvious data is repeating. Please make the discussion more comprehensive.

13.  All tables are consistent instead table 2. Correct.

14.  Check Figure 8 title. It is unclear for (b)

Author Response

Specific comments:

Point 1: Authors are advised to check the whole manuscript for typograph errors. Example: some of the sentences at the end have a space before point. Page 1, line 33; 38;46, page 2, line 48; 64;94, page 6, line 179; 186;189;194;197;201;204; 209; 212.......

Reply: correction done, please.

Point 2: Some of the abbreviation are not defined for the first time.

Reply: Abbreviations are checked and updated, please.

Point 3: Page 1, line 43:“Policies on SWM seem satisfactory, and there is a need for implementation“. If they seem satisfactory, why is then a need for implementation?

Reply: It is updated in lines 69-71 as per the comment, please.

Point 4: Page 2 line 72:“The uncertainties observed on the part of policymakers for further capital investment in the sector because of lessons learned during the last decade.“ Unclear? Revise.

Reply: The sentence has been revised and clarified in lines 106-108, please.

Point 5: Table 1. Align text and labels

Reply: correction done please.

Point 6: Authors are advised to centre all formulas/Equations and to crossref. with the text. Also attention should be directed to equation 3. There should be a large bracket at start and before:1000, in the situation as it is written it means that only the last bracket: (Qty*kg/d(g)xSP(kg)) is divided by 1000. As I understood all brackets need to me calculated as SUM, then divided by 1000 to get the result in recyclables per ton.

Reply: very valid observation by the expert in the SWM field. The authors really appreciate it. Large brackets are inserted and crossref added to equations, please.

Point 7: Page 6, line 182 authors write: paper 6%, and if we look at the Appendix for paper is 8.36% stated. Which one is correct? Please check.

Reply: as per the suggestion of one of the reviewers, graphs are enough to depict the results followed by details as reflected in appendix 1. Correction made accordingly. 8.36% paper figure is correct in the case of Karachi. Thanks a lot for the correction, please. 

Point 8: Authors should uniformly state results for same parameter. Example: 6%, 1,8%, 60.13%....same parameter (%), but with different decimal places??

Reply: compliance ensured.

Point 9: Numbers are sometimes written by point and sometimes by comma. Correct, to be uniformly.

Reply; compliance ensured, please.

Point 10: Figures should also be consistent, with same style. Also the text should be same as in other parts of MS. Times New Roman.

Reply: Compliance ensured with thanks, please.

Point 11: Re-check page 7, line 223-226 with Appendix values. Line 236- Peshawaer (45%) is not visible in the Appendix section?

Reply: There was a formatting/ page setting issue that is corrected.

Point 12: Some sentences have the same meaning, which means that some obvious data is repeating. Please make the discussion more comprehensive.

Reply: The authors tried their best and updated the discussion section and added a new paragraph to make it more comprehensive.

Point 13: All tables are consistent instead table 2. Correct.

Reply: correction done, please.

Point 14: Check Figure 8 title. It is unclear for (b)

Reply: correction done, please.

Reviewer 4 Report

This paper touch important issues related to waste management in Pakistani where the system is far behind the developed countries. However, this paper is prepared which high negligence. There are not references included across the whole manuscript which makes this manuscript hard for reading.
This paper is overload with repetitive data and information. The graphs are described too thoroughly by text. In the text it should be stated only the critical information which are stated in the graphs or tables.
Because of lack of references in the body text it is impossible to evaluate the discussion section. In the discussion section there is not any reference to other authors. The conclusion section need be shortened with only the most relevant information related with scope of the research described in this paper.

Additional comments are included in the attached file.

Author Response

Point 1: However, this paper is prepared which high negligence. There are not references included across the whole manuscript which makes this manuscript hard for reading.

Reply: Thanks for thoroughly reviewing the manuscript. The authors take your words as an opportunity to refine the manuscript. Citations of references are mentioned within the text properly. Previously, these were missed due to some software error when the document converted from word to PDF format.

Point 2: This paper is overload with repetitive data and information.

Reply: manuscript updated accordingly as per comments.

Point 3: The graphs are described too thoroughly by text. In the text, it should be stated only the critical information which is stated in the graphs or tables.

Reply: the text is summarized as per kind suggestions.

Point 4: Because of the lack of references in the body text it is impossible to evaluate the discussion section.

Reply: Citations of references are mentioned within the text properly.

Point 5: In the discussion section there is not any reference to other authors.

Reply: Compliance ensured, please.

Point 6: The conclusion section need be shortened with only the most relevant information related with scope of the research described in this paper.

Reply: The authors tried their best to shorten the conclusion section as per kind observations.

Point 7: Additional comments are included in the attached file.

Reply: In-depth review is highly appreciated. Amendments were made accordingly.

  1. Abbreviations checked in manuscript
  2. Treatment options for organic waste elaborated
  3. Text summarized in the “background information and key waste-related data” section
  4. Legend/ key Note is added after Figure 5 for clarification to readers for radar diagrams.
  5. Cited reference for emission inventories.
  6. Table 5 was removed as per comments. https://www.iges.or.jp/en/search-result?search_api_fulltext_=calculator&items_per_page=10 . The emission calculator is available at the link.
  7. Table 6 was removed as per the comment.
  8. Figure 11 was also removed.

As per observations in the Reviewer Table; the authors tries best to improve the crossed points.  

Reviewer 5 Report

Paper is written well. However, I have minor suggestions.

Improve abstract by including theoretical background of concept and limitation of study. 

Mention research gaps in literature review section.

Conclusion section should contains more future research directions.

Author Response

Point 1: Improve abstract by including theoretical background of concept and limitation of study. 

Reply: Abstract section is updated as per kind observation. Thanks for the suggestion.

Point 2: Mention research gaps in literature review section.

Reply: The section is updated by incorporating the observations and can be viewed in track change.

Point 3: Conclusion section should contain more future research directions.

Reply: future research direction added in the conclusion section please (last bullet point).

As per observations in the Reviewer Table; the authors tries best to improve the crossed points.